# Spatial Speaker ID: Joint Spatial and Semantic Learning for Multi-Microphone Speaker Identification on Short Far-Field Utterances

## Abstract

Speaker identification is the task of identifying a person who is currently talking by analysing microphone signals. Typical automatic speaker identification systems use a single microphone and require complete utterances of 4-10 seconds in length to accurately identify a person from an enrollment set. We introduce the related problem of detecting which person is talking among several people in a room when the utterances are very short, e.g., a single word, or a short laugh. Since utterance lengths are too short for conventional methods, we take inspiration from the way humans solve this problems - using two ears and a joint understanding of both semantic and spatial context. To solve this problem, we propose Spatial Speaker ID, which uses banded covariance features derived from multi-microphone input along with conventional banded power to identify talkers based on both the semantic characteristics of a sound and the spatial location of a sound. The internal representation learnt in Spatial Speaker ID jointly contains both spatial and voice characteristic information and is learnt contrastively, whereby two utterances that come from the same talker in the same location are required to have similar embeddings. We learn a binary classification downstream task that determines if two sets of embeddings come from the same talker in the same location. Using this binary classifier, we compare multiple ways of presenting the microphone covariance features to the upstream models. We show the importance of spatial information for identifying talkers on short utterances with interfering noise.

## 1 Introduction

Most acoustic speaker identification systems use a single channel of input. This works well when there is only person talking at once, the signal-to-noise ratio (SNR) is favourable and the person to be identified talks for long enough to distinguish themselves from the other candidates. In this work, we deal with a case in which multiple people are present in a room but say only very short utterances, often simultaneously, in noisy, reverberant far-field conditions. To deal with this challenging case we turn to multiple microphone input channels to disambiguate who is speaking.

Humans deal with these sorts of complex acoustic conditions by forming joint spatial and semantic content. That is, we not only use the sound of a person's voice to determine who spoke, but also complex acoustic features from our two ears, such as interaural level difference (ILD) and interaural time difference (ITD). Inspired by this, we propose a model, Spatial Speaker ID, with multiple microphones in first order ambisonic (FOA) format to identify who is currently speaking.

To train our model, we create assemble a dataset of short utterances (0.3-2 s in length) and convolve them with the output of a FOA room impulse response (RIR) generator which can accurately simulate the complex spatial signatures of speech incident on a microphone array from different locations in the same room. We use this training data in a contrastive representation learning regime which requires that our model must jointly distinguish between locations and talkers. The resulting latent space contains information not only of who spoke, but where they spoke. We evaluate the effectiveness of this upstream model with several downstream tasks and we find that the spatial information has a dramatic positive influence on the system's ability to determine who is talking.

## 1.1 Multi-Microphone Speaker Identification on Short Far-Field Utterances

This work is motivated by the following real-world problem statement. A room contains multiple people. People may leave or enter the room at any time. People may move from place to place within the room at any time. From time to time people speak short utterances such as a single word, an exclamation, a short laugh, a cry of surprise. Sometimes these utterances partially overlap or occur simultaneously. There is background noise present, such as road noise from outside or the sound of cooking. The sound in the room is recorded by a small aperture multi-microphone array placed approximately 1-5m from the people. We want to detect the short utterances and also identify which person spoke each utterance. We call this problem Multi-Microphone Speaker Identification on Short Far-Field Utterances (MSISFU).

This problem has important applications in engagement detection and analysis, interactive gaming, Extended Reality (XR) and content personalisation for social scenarios in which several people are engaged together in a media experience, such as watching a football game together in a living room. In this example, the microphone array might be mounted on the television, coffee table or credenza, multiple people might be seated on the couch, while others might be at the refridgerator getting drinks.

This problem has several similarities with conventional Speaker Identification and Verification. We can use features of each person's voice in order to determine who is speaking. However, unlike the conventional Speaker Identification and Verification problem statement, we consider utterances as short as 300ms, and they may occur simultaneously or partially overlapped. Since this is a far-field acoustic environment, acoustic reverberation within the room plays a major role. Therefore, the problem also bears similarities to the cocktail party problem. Since a multi-microphone array is available, we are inspired by the way the human brain uses spatial information to disentangle sound in real-world reverberent acoustic environments.

## 1.2 Joint Semantic and Spatial Representation Learning

To solve the problem of detecting who is speaking on short far-field utterances, we combine contrastive representation learning and spatial input and develop training technique which enforces a latent space containing both semantic (speaker identification) and spatial information. Our contrastive proxy task requires that embedding pairs are similar for input pairs containing the same speaker in the same location in the same room speaking different short utterances, but more distant for input pairs containing different speakers or for input pairs containing the same speaker in different locations in the same room.

A key question to tackle is how should multiple microphone channels be presented to a model in order to obtain best results. A naive approach would be to derive a single direction-of-arrival estimate from the multiple microphones. However, these approaches have proven error-prone in real world reverberant acoustic environments where the direction of arrival is obscured by reflections from different directions. We therefore use the spatial covariance matrix of the microphone array in log-spaced frequency bands as the core input features for our system. We compare multiple alternatives for representing the covariance information to our model, using the singular value decomposition technique from SALSA (Nguyen et al. (2022b)) as a baseline, but also consider a variety of cheaper fixed and learnt alternatives. In particular, we introduce a method of summarising covariance which we call the Power Vector. We show that the Power Vector provides very efficient learning of spatial information by the network, while remaining extremely simple and cheap to implement.

## 1.3 Summary

In summary, this work contains three new contributions:

1. We introduce a new real-world task, Multi-Microphone Speaker Identification on Short Far-Field Utterances (MSISFU), with applications in engagement detection and analysis, interactive gaming, Extended Reality (XR) and content personalisation.

2. We propose a new contrastive learning approach that learns a joint spatial and semantic (speaker identification) representation from first order ambisonic (FOA) input. We show

that these learnt representations are highly effective on the MSISFU task along with related downstream tasks.

3. We describe and compare a family of techniques for presenting banded spatial audio information to a machine learning model. One member of the family, the Power Vector technique, stands out as providing excellent results compared to recent techniques such as SALSA while remaining extremely efficient and simple to implement.

To our knowledge this is the first work that applies contrastive learning and spatial input to a speaker identification task.

## 2 RELATED WORK

### 2.1 SPEAKER IDENTIFICATION AND VERIFICATION

Speaker Identification and Verification are long-standing research tasks with a rich history of contributions. Speaker Identification is the task of determining, from a recording of one person's speech, which person from an enrolled set is speaking. Typical solutions require 3-10s of near field speech in relatively noise-free conditions from a single microphone in order to determine who is speaking. Speaker Identification has applications in personalisation systems, particularly in virtual assistants, where the context of who is speaking is used to better understand the command. For example, in order to "play my music", a virtual assistant must recognise who I am.

Speaker Verification is a related task that starts with a presumed identity of the speaker and produces a confidence that the person speaking is who they say they are. Typically the recording of the speaker's voice is from a single near-field microphone, relatively free of noise and at least 3-10s in length. Speaker Verification has applications in security, such as protecting access to a bank account or door lock.

As summarised by Kinnunen & Li (2010) and Hansen & Hasan (2015), Speaker Identification and Verfication approaches have commonly used a "supervector" which encapsulates statistical differences in the distributions of features present in the speech of a particular speaker compared to those in a Universal Background Model (UBM). More recent approaches, such as Desplanques et al. (2020), use a deep learning based representation space known as an x-vector to distinguish speakers. In the last few years, self-supervised speech representation learning models such as WavLM (Chen et al. (2022)) produce SOTA results on the downstream task of Speaker Verification when evaluated on datasets such as VoxCeleb, which has a minimum utterance length of 4s and a mean utterance length of 8.2s (Nagrani et al. (2017)). On the other hand, our MSISFU task requires 0.3-2s length utterances.

Conventional Speaker Identification and Verification systems are generally designed to work with only a single input channel. When multiple microphones are present on the device, which is typical in smart speaker and automobile microphone arrays, the Speaker Identification and Verification system is often preceeded by a beamformer, which combines microphone signals together to form the cleanest possible monophonic signal on which to identify speech. However, the core speech processing model is trained on single channel input and operates without any directional context, or context on how loud any interferring speech or noise sources were.

### 2.2 SOUND EVENT LOCALISATION AND DETECTION (SELD)

In recent years, it's been shown that machine-learning models can learn semantic information through self-supervised learning with proxy tasks. For example, HuBERT (Hsu et al. (2021)) learns a latent space from single channel input that contains semantic (phonetic and speaker identification) information. Recent works have shown that spatial information can be used for the task of Sound Event Detection and Localisation (SELD).

There are a range of spatial feature extractors that are being used in the SELD literature, such as, FOA intensity vectors (Yasuda et al. (2020)), SALSA (Nguyen et al. (2022b)), GCC-PHAT (Cao et al. (2019)), SALSA-Lite (Nguyen et al. (2022a)) or a combination of these frontends (Wang et al. (2020)). The room impulse responses (RIR) generator we used in this work generates FOA RIRs.

As such we compare against SALSA, a state-of-the-art feature extraction frontend for spatial audio machine learning models.

Our problem statement bears similarities with previous work on Sound Event Localisation and Detection (SELD). In both cases, the sounds in question might be short. In our case, the sounds of interest are human utterances and the task is to detect which person produced each sound, wheres in the SELD case the task is to detect, localise and classify each sound into one of several classes (eg a dog barking, a car engine starting). Note that in our problem, we do not necessarily need to localise each person in polar or cartesian coordinates. It is only necessary to disambiguate them when they speak.

## 3 METHOD

### 3.1 SPATIAL SPEAKER ID PROXY TASK

Our primary Spatial Speaker ID model is trained on a contrastive proxy task that encourages the formation of a latent space that jointly represents semantic (what does the voice sound like?) and spatial (where does it come from?) information. The model should ignore interfering noise. The inputs to this proxy task are 2.5 second long simulated FOA signals of a single speaker $S_a$ speaking a short utterance $U_b$ at location $L_c$ in a reverberant room $R_c$ with interfering noise $N_e$ present at location $L_f$. We form triplets of reference, positive and negative examples and apply triplet loss in order to encourage the following behaviours.

- Two unique utterances $U_1$, $U_2$ from the identical speaker $S_1$ in the identical location $L_1$ in room $R_1$ should produce near embeddings.
- Two unique utterances $U_3$, $U_4$ from the identical speaker $S_2$ in two unique locations $L_2$, $L_3$ in room $R_2$ should produce distant embeddings.
- Two unique speakers $S_3$, $S_4$ in the identical location $L_4$ in room $R_3$ should produce distant embeddings, regardless of the utterances.
- The model should be robust to interfering noise and level differences. i.e. two utterance pairs should produce similar embedding differences regardless of the level of the utterances $U_a$, the content of the interfering noises $N_1$, $N_2$, the location of the interfering noises $L_5$, $L_6$ and the signal-to-noise ratio (SNR).

A corollary of the above is that two unique speakers $S_5$, $S_6$ in two unique locations $L_7$, $L_8$ should also produce distant embeddings. Note that we never compare signals from distinct simulated rooms $R_4$, $R_5$ since the MSISFU task is about understanding who is talking within a single acoustic environment.

To encourage noise robustness we mix a distinct noise signal simulated from a distinct location at a distinct SNR with each utterance. To encourage level independence we apply a random gain to each simulated signal before supplying it to the model.

Our Spatial Speaker ID model provides an embedding vector for each input frame of audio. We therefore summarise the output embeddings over time using a trained self-attention pooling layer before applying the triplet loss described above. In addition to these behaviours we also want the model to know when speech is actually present since we want to be sensitive to very short utterances (as short as 0.3s within our training inputs of length 2.5s). We therefore apply an additional mean-square loss that requires null output embeddings (i.e. embeddings with a very small magnitude in every dimension) for frames of input that contain noise only without speech.

### 3.2 SIMULATED AMBISONIC ROOM IMPULSE RESPONSES

To simulate utterances and noise from locations in reverberant rooms, we use an in-house first order ambisonic room impulse reponse (RIR) simulator. Our data sources of speech and noise are monophonic and are convolved with these RIRs to create simulated ambisonic signals that can be mixed together as described above. Our RIR simulator has the following input parameters:

- The position of the source relative to the simulated ambisonic microphone expressed as cartesian coordinates in metres.

- The reverberation time ($T_{60}$) of the simulated room in seconds in multiple frequency bands, which affects how quickly the simulated RIR decays.

- The volume of the room in $m^3$, which affects the density of reflections in the simulated RIR.

This parametrisation allows us to simulate sources at multiple locations in an identical room. It also allows us to simulate as many unique rooms as desired. Our RIR simulator is implemented in Pytorch and can operate efficiently on GPU, which means that we can efficiently draw from an infinite set of simulated rooms online during training to prevent overfitting to a particular set of acoustic environments.

### 3.3 FEATURE EXTRACTION

The input to the Spatial Speaker ID model is an FOA signal. We use a fixed feature extraction pipeline at the model input which applies short time Fourier transform (STFT) with a raised-sine analysis window, 20ms step size with a 50% overlap and a half-frequency-bin kernel shift to analysis each of the FOA microphone channels W, X, Y and Z. This results in complex values for each frequency bin of each channel. We then compute the 4x4 complex spatial covariance matrix across channels in each frequency bin and apply smoothing in time and frequency.

We use a Mel-like frequency smoothing operation which summarises the STFT frequency bins into bands occupying approximately equal log spectrum. When operating on 16kHz-sampled input, this results in 48 log-spaced bands every 20ms. For each of these bands we have a 4x4 complex spatial covariance matrix. These bands are then further smoothed in time across frames to ensure approximately equal statistical power in the spatial covariance matrix for each band.

These unnormalised banded complex spatial covariance matrices are then reduced to real vectors as described below.

### 3.4 REPRESENTING MULTI-CHANNEL SIGNAL POWER AS A VECTOR

The power spectrum of a monophonic signal simply describes each time-frequency tile in terms of a scalar signal-power quantity (or some function of the signal-power, such as the logarithm).

When dealing with an $N$-channel signal, the natural extension of signal-power is the $N \times N$ covariance, and it is worth noting that the covariance matrix is a non-negative definite (Hermitian) matrix, and when multiple uncorrelated sound sources are captured by a microphone array, the covariance of the captured signals will be the sum of the covariance contributions of each individual source (superposition).

It is convenient to separate a covariance matrix, $C$, into total power ($P_{tot} = \text{tr}(C)$) and a unit-trace matrix $C'$, so that: $C = P_{tot}C'$. In general, at this point $C'$ is a complex unit-trace matrix. It is desirable to extract a real vector $P_{cov}$ from this matrix $C'$ in order to present it as a feature to a machine learning model. We do this by applying a covariance mapping function $M$:

$$P_{cov} = M(C') \tag{1}$$

We consider multiple alternate implementations of $M$:

- ($M_0$): Drop all covariance information,

- ($M_{learnt}$): concatenate and flatten the real and imaginary parts of $C'$ and present them through a learnt linear projection,

- ($M_{UT}$): since $C'$ is Hermetian, take its upper triangle, concatenate and flatten the real and imaginary parts, and let this function be,

- ($M_{PV}$): use the Power Vector $P_{dir}$ which is introduced below, or

- ($M_{SALSA}$): use the singular value decomposition technique described in SALSA (Nguyen et al. (2022b)).

## 3.5 POWER VECTOR

For convenience, the $N \times N$ matrix $C'$ may be re-shaped into a column vector and mapped via a unitary matrix, $F$, to form:

$$P_E = F \times vec(C')$$ (2)

Two benefits derive from this reformulation. Firstly, the vector $P_E$, is easier to manipulate than the matrix $C'$. Secondly, by judicious choice of the unitary matrix $F$, it is possible to ensure that $P_E$ contains only real values, and the first element of $P_E$ is a constant (equal to $\frac{1}{\sqrt{N}}$).

Since $\{P_E\}_1 = \frac{1}{\sqrt{N}}$, we can omit this element, and form a new column vector (of length $N^2 - 1$), which we call the "power vector":

$$P_{dir} = \sqrt{\frac{N}{N-1}}\{P_E\}_{2:N^2}$$ (3)

We have now formed a one-to-one mapping between a covariance matrix, $C$, and an alternative representation: $\{P_{tot}, P_{dir}\}$, where scalar $P_{tot}$ describes the total power of the $N$-channel signal, and the $N^2 - 1$ length column vector $P_{dir}$ describes the directional aspects of the $N$-channel signal.

$P_{dir}$ has the following useful properties:

1. $||P_{dir}||_2 = 0$ tells us that the $N$-channel microphone signals are uncorrelated and of equal amplitude
2. $||P_{dir}||_2 = 1$ tells us that the $N$-channel microphone signals are linearly dependent, so that could be formed by panning a monophonic signal into the $N$ channels
3. Given an $N$-channel signal that was formed by panning a monophonic signal with a gain $N$-vector $G$ (and where $||G||_2 = 1$), then $P_{dir} = \sqrt{\frac{N}{N-1}}\{F \times vec(G \times G^*)\}_{2:N^2}$
4. Given two gain vectors, $G_X$ and $G_Y$ (where $||G_X||_2 = ||G_Y||_2 = 1$), we define the "similarity" of two gain vectors as $S_{XY} = |G_X^* \times G_Y|$. Then: $P_{X,dir}^T \times P_{Y,dir} = \frac{NS_{XY}^2 - 1}{N-1}$

The last item above tells that the similarity of two gain vectors (the degree to which two sounds, as they are panned into the $N$-channel signal, are difficult to separate) is directly mapped to the dot-product of their respective power-vectors.

## 3.6 DOWNSTREAM TASKS

To evaluate the performance of our primary Spatial Speaker ID model, we define several downstream tasks that operate on the Spatial Speaker ID embedding.

- **Binary Spatial Speaker ID (B-MSISFU)**: We present a pair of input vectors (simulated as for the upstream task) and classify whether those input vectors are from identical talkers in an identical location or otherwise. We apply binary cross entropy loss during training. Performance on this task is our primary indicator that the latent space is appropriate for the MSISFU task.
- **Direction-of-Arrival (DoA)**: We present an input vector (simulated as for the upstream task) and predict the direction-of-arrival (a 3-dimensional unit vector in Cartesian coordinates). This task measures how the latent space disambiguates acoustic locations.
- **Speaker Identification (SID)**: We present an input vector (simulated as for the upstream task) and classify which speaker in the training set spoke the utterance. We apply cross-entropy loss during training. This task measures how the latent space encapsulates the speaker's identity.

We implement each of these downstream tasks by applying self-attention pooling to summarise the framewise Spatial Speaker ID embeddings over time and then using a simple two-layer MLP of appropriate dimensionality for each task.

# 4 EXPERIMENTS

## 4.1 OVERVIEW

In this work we perform three experiments:

1. Experiment 1 was designed to prove the basic premise that spatial information is useful for speaker identification. To do this we train a monophonic model using $M_0$ and a spatial model trained using $M_{PV}$.

2. Experiment 2 sought to compare how the amount of covariance information supplied to the upstream model affects performance on the MSISFU task. To acheive this, we train models using learnt covariance mappers $M_{learnt}$ with various sized bottlenecked outputs, 1, 2, 4, 8, and 16.

3. Experiment 3 was created as a baseline comparison against a state-of-the-art FOA frontend. We compare our linear covariance mappers with a SALSA inspired covariance mapper $M_{SALSA}$.

All of these experiments have the upstream models trained for 50,000 steps. Subsequently the upstream and downstream paths were trained for an additional 150,000, 100,000 and 10,000 steps for experiments 1, 2, and 3, respectively. The number of talkers used in each batch was 32 for the first experiment and 14 for the last two experiments.

## 4.2 MODEL ARCHITECTURE

The upstream portion of our model is composed of a linear projection, convolutional layers followed by a three-layer transformer encoder as described by Vaswani et al. (2017) with a hidden size of 768, 8 heads.

Each downstream task is learnt from the framewise Spatial Speaker ID embeddings from the final layer of the transformer encoder. We apply self-attention pooling Safari et al. (2020) to summarise the framewise embeddings into a single embedding. Our self-attention pooling has an additional linear layer of hidden size 768 and ReLU activation in the weighting. We then apply an appropriately-sized MLP, with ReLU activation and intermediate LayerNorm.

In some experiments we allow our upstream model to continue training while the downstream tasks train. However, we ensure that gradients from the downstream tasks do not backpropagate into the upstream model so that the models are effectively trained independently.

Self-Attention Encoding and Pooling for Speaker Recognition but with an additional

## 4.3 DATA PREPARATION

Preparing data for the spatial talker identification task requires careful construction of batches and data augmentation. Every batch requires multiple utterances from each talker, utterances to be short, interference noises and the spatialisation of sounds with matching locations between different utterances.

We produce audio training samples that are 2.5 seconds in length, consisting of interfering noise with a short utterance of length 0.3-2 seconds that appears at a random time. These audio samples form batches, where each batch is created by selecting 30 random speech segments for each of the random speakers. In experiment 1, we selected 14 random speakers per batch. In experiments 2 and 3, we selected 32 speakers per batch.

We use a voice activity detector (VAD) to ensure that the short segments we cut from the original training corpus actually contain speech. To avoid abrupt starts and ends of our short utterances, we apply a 50ms fade-in and fade-out.

At each batch we draw a random selection of interference noise to mix with the speech. We discard noise samples from the training corpus that are shorter than 2.5 seconds in length. To ensure the model is well-behaved when no speech is present, we also arrange for 10% of the training samples in each batch to contain only noise without any speech present. These samples are considered to

have no speaker and when forming triplets are negative samples for a reference containing a speaker, or positive samples for a reference without a speaker. This helps to enforce that the null (no-speech) embedding is distinct from all speech embeddings.

We draw simulated speech locations from around the microphone with horizontal distance $h_t \sim \mathcal{N}(1.5, 0.5)$m, vertical distance $v_t \sim \mathcal{N}(0.5, 0.3)$m. The azimuth direction of arrival is initially sampled uniformly $\theta_0 \sim \mathcal{U}(0°, 360°)$. For the noises, we draw distances further away from the microphone with settings, $h_n \sim \mathcal{N}(5, 2)$m, $v_n \sim \mathcal{N}(1, \frac{1}{\sqrt{2}})$m. To make the comparison of spatial locations sensible and realistic, for each batch we sample one set of room acoustics. We sample $T_{60}$ from $r \sim \mathcal{N}(0.3, 0.1)$s and room dimensions from length $l \sim \mathcal{U}(3, 6)$m, width $w \sim \mathcal{U}(2, 5)$m and height $h \sim \mathcal{U}(3, 4)$m.

Two additional constraints are applied to the simulated placement of speakers to make the proxy task sufficiently difficult and realistic.

- When simulating two unique utterances $U_3$, $U_4$ from the identical speaker $S_2$ in two unique locations $L_2$, $L_3$, we apply additional logic to ensure that $L_3$ and $L_2$ differ by at least $6°$ but no more than $30°$ in azimuth. This ensures the proxy task is sufficiently difficult that the model does not trivially rely only on spatial information.
- We add a small amount of random noise (uniformly distributed over a sphere with 30mm radius) to all selected speaker locations. This ensures that any two locations are never truely identical and ensures our model is robust to movement over time that would be expected from real humans in real rooms.

The interfering noise is mixed with the speech to achieve a randomly selected signal-to-noise (SNR) ratio uniformly distributed between -5dB to +20dB. After mixing, we apply a random gain uniformly distributed between -30dB and +30dB to each training sample to encourage level independence.

### 4.4 LOSS FUNCTION

To train our Spatial Speaker ID network contrastively, we used a triplet margin loss Hoffer & Ailon (2015). This approach requires sampling triplets from the training data, where triplets are made up of an anchor, positive and negative embedding. Using all possible triplets from the training samples would be impractical given the number of triplets scales quadratically with the number of talkers and cubically with the number utterances per talker in a batch, given our batch construction. With the settings we used, the number of possible triplets would be in the order of millions for our 14 talkers per batch experiments. To reduce the number of triplets, we had a maximum of 100 triplets per anchor, where each training sample could act as an anchor.

There also needs to be a well distributed selection of the Spatial Talker ID task cases, as listed in the Spatial Talker ID Proxy Task section. To achieve this we parameterised the sampling of negatives. Each triplet had a probability $p$ of selecting a one of the two more difficult MSISFU cases, an identical speaker $S_2$ in two locations $L_2$, $L_3$, and two speakers $S_3$, $S_4$ in an identical location $L_4$. The higher proportion of negatives sampled from difficult cases, created a more challenging task to train on, thus robustifying the Spatial Talker ID model. Through experimentation, we found that setting $p$ to 25% provided a good balance for learning a representation that contains both spatial and voice characteristic information.

To further regularize the models, we introduce an additional loss term for the null embeddings. This loss, based on Mean Squared Error (MSE), encourages all null embeddings to be as close to zero as possible.

### 4.5 DATASETS

We used LibriSpeech Panayotov et al. (2015) train-other-500 subset for our experiments. To perform a valid evaluation of downstream tasks such as Speaker Identification (SID), unseen utterances are required from the speakers seen during training. Investigating the performance of Spatial Speaker ID models on unseen talkers is task for future works.

For validation, we used 5% of the dataset, which is approximately 25 hours. We ensured that each speaker in the dataset had at least $\lfloor \frac{|U| \times 0.05}{|N|} \rfloor$ utterances in the validation set, where $|U|$ is the total

Table 1: Experiment 1

| Frontend | B-MSISFU % ↑ |
|----------|--------------|
| Mono | 79.0 |
| Power Vector | **92.4** |

Table 2: Experiment 2

| Covariance Mapper | B-MSISFU % ↑ | DoA Error ° ↓ | SID % ↑ |
|-------------------|--------------|---------------|---------|
| Learnt 1 | 86.9 | $90.2 \pm 66.6$ | **30.8** |
| Learnt 2 | 85.9 | $88.3 \pm 70.3$ | 30.2 |
| Learnt 4 | 89.1 | $84.6 \pm 72.6$ | 26.9 |
| Learnt 8 | 89.7 | $88.7 \pm 67.1$ | 25.9 |
| Learnt 16 | 90.8 | $85.4 \pm 75.8$ | 22.5 |
| Upper Triangle | 90.3 | $88.8 \pm 68.6$ | 24.6 |
| Power Vector | **91.1** | $\mathbf{4.3 \pm 7.3}$ | 17.0 |

number of utterances in the dataset and $|N|$ is the number of speakers. We also took measures to make the validation set had the same distribution of utterance lengths as there is in the whole dataset.

For interference noise, we used AudioSet balanced train segments for training and AudioSet Gemmeke et al. (2017) eval segments for validation.

## 5 RESULTS AND DISCUSSION

In Table 1, the reported B-MSISFU accuracy is calculated using only the cases with identical speaker $S_1$ and location $L_1$, and different speakers $S_1$, $S_2$ and locations $L_1$, $L_2$. The Power Vector frontend shows an excellent 13.4% improvement in performance over the Mono model. These results reveal the importance of spatial information for identifying speakers on short utterances in far-field scenarios with interfering noise.

In the following two experiments all of the four MSISFU task cases are included in the B-MSISFU accuracy.

In Table 2, the 'Learnt $d$' experiments refer to covariance mappers $M_{learnt}$ that have a learnt linear projection, with an output dimension size of $d$. These results reveal that utilising more spatial information improves the performance on B-MSISFU. Observe, that although reasonable linear mappings $M_{learnt}$ are possible to learn, the hand-crafted PV mapping $M_{PV}$ still outperforms them. Interestingly, only the upstream model trained with $M_{PV}$ was able to have reasonable direction of arrival information derived from its learnt representation. This could be in part to the numerical stability and lack of redundancy in the PV features, which may affect the shape of the latent space. This is also likely a side effect of the MSISFU task too, where physical locations are not required to be estimated. The model simply needs to represent different locations in different positions in the latent space. By observing the SID task, one can see how models trained with less spatial information, such as 'Learnt 1', contain more speaker voice characteristics in their embeddings. This reveals that the talker information is much less important if spatial information is available.

In Table 3, we compare the B-MSISFU classification accuracy of our covariance mappers with the SALSA inspired SVD covariance mapper $M_{SALSA}$. Note, how there are 16 unique values in the upper triangle of a Hermitian matrix. As such, any linear projection-based covariance mapper with an output of 16 is able to retain all covariance information. Now observe, the performance of linear projection-based covariance mappers that retain full covariance information. These include Learnt 16, Upper Triangle, and Power Vector covariance mappers. These methods perform comparably to the more computationally expensive technique, SALSA. To train the model using $M_{SALSA}$ for this experiment took 3.8 days, whereas all of the other model training times for the same number of steps

Table 3: Experiment 3

| Covariance Mapper | B-MSISFU % ↑ |
|---|---|
| Learnt 1 | 83.2 |
| Learnt 2 | 83.1 |
| Learnt 4 | 85.0 |
| Learnt 8 | 86.4 |
| Learnt 16 | 88.4 |
| Upper Triangle | 88.3 |
| Power Vector | 88.3 |
| SALSA | **89.7** |

took approximately 1.6 days. The SALSA-inspired covariance mapper $M_{SALSA}$ only performs approximately 1.4% better on the B-MSISFU task, despite its significantly higher computational cost.

## 6 CONCLUSION

We introduced the Multi-Microphone Speaker Identification on Short Far-Field Utterances (MS-ISFU) task. The MSISFU task enables the learning of a latent space containing both voice and spatial characteristic information. Moreover, the task showcases a more realistic alternative to the traditional machine learning task Speaker Identification, which uses longer utterances on monophonic audio. This paper details how MSISFU can be learnt contrastively using a proxy downstream task. We described and compared a family of techniques for transforming covariance matrices into suitable inputs for machine learning models, including our proposed Power Vector technique.

We conducted three experiments. Experiment 1 proved that the introduction of spatial information was extremely useful for the MSISFU task, boosting the performance on the B-MSISFU metric by 13.4%. The results of Experiment 2 demonstrate that model with full covariance information, attain better results. Additionally, we observe beneficial properties in the upstream model's latent space when training it with hand crafted Power Vector features over a learnt linear projection covariance mapper. The Power Vector features allowed the model to learn a latent space which contains DoA information. Experiment 3 showed that linear projection based covariance mappers have small degradation in performance (1.4%), showing comparable results to the more computationally expensive SALSA inspired method. Our evaluation demonstrated that models trained on the MSISFU task can effectively disambiguate speakers based on both their voice and spatial location.

There are many avenues for future work:

- Investigating a Spatial Speaker ID model that can handle generating multiple tracks of embeddings, such that multiple speakers can be identified in a single utterance or per time frame embedding. A model of this type could be trained with a permutation invariant training loss.

- Researching if it is possible to improve MSISFU results further by allowing the downstream tasks to learn from the embeddings of each layer in the upstream model, following the likes of Hsu et al. (2021), Chen et al. (2022) and Schneider et al. (2019).

- Constructing a more comprehensive evaluation of frontend representations and their performance on the MSISFU task.

ACKNOWLEDGMENTS

The acknowledgments have been anonymised, as the paper is under double-blind review.

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
