# OpenReview forum: "Spatial Speaker ID: Joint Spatial and Semantic Learning for Multi-Microphone Speaker Identification on Short Far-Field Utterances"
_ICLR.cc/2025/Conference — ICLR 2025 Conference Withdrawn Submission_

### Official Review · Reviewer_ksR5 · 2024-10-27

**Soundness:** 3
**Presentation:** 1
**Contribution:** 2
**Rating:** 1
**Confidence:** 4

**Summary:**

This paper presents a new task by extending conventional speaker identification task with sound event detection to handle real-world speaker identification problem. It aims at designing a unified framework to handle far-field speaker identification&verification via explicitly encode the multi-channel power as vector representations.

**Strengths:**

1. The novelty of the paper is quite clear and the proposed frontend is technically sound.
2. The data preparation process is clearly explained.

**Weaknesses:**

1. This presentability of this paper is extremely poor. For example, there is no figure in the paper; the section 4.2 has a sentence not supposed to be in the paper; too many terms apart from data collection are not clearly explained with randomness; Table 1 and 2's caption is not there. The presentation quality of this paper does not fit with the conference at all.
2. The definition of task and linking to real-world scenario really confuses the reviewer. First, the reviewer has trouble understand the difference between the described task in this paper and speaker diarization with conversation scenario; Second, the "too-short" scenario seems do not fit very well with real-time conversations, which shall be continuously, decently long.
3. Following 2), the scenario is simulated while there are datasets such as HI-MIA, which is recorded in the real-time environment with noise setup and microphone array [1]. I suggest to have a check on that and benchmark at least speaker identification on it.
4. Following 3), the motivation to use LibriSpeech, a reading speech dataset, is not valid from reviewer's point of view.

[1] X. Qin, H. Bu and M. Li, "HI-MIA: A Far-Field Text-Dependent Speaker Verification Database and the Baselines," ICASSP 2020 - 2020 IEEE International Conference on Acoustics, Speech and Signal Processing (ICASSP), Barcelona, Spain, 2020, pp. 7609-7613, doi: 10.1109/ICASSP40776.2020.9054423. keywords: {Training;Databases;Speech enhancement;Acoustics;Microphone arrays;Task analysis;Testing;open source database;text-dependent;multichannel;far-field;speaker verification},

**Questions:**

The questions has been listed in weakness part.

**Details Of Ethics Concerns:**

The reviewer thinks there shall have no ethical concerns in this paper.

---

### Official Review · Reviewer_yHHn · 2024-11-03

**Soundness:** 2
**Presentation:** 2
**Contribution:** 1
**Rating:** 1
**Confidence:** 4

**Summary:**

This paper focuses on extracting an embedding that considers both spatial and spectral context from a very short utterance. The authors emphasize the importance of such scenarios (called MSISFU). The main idea is to focus on the spatial covariance in the ambisonic microphone scenarios and represent the proposed power vector from the spatial covariance fed to a simple neural network to perform speaker recognition. The experiments are based on their in-house ambisonic data simulation applied to the Librispeech data, and training and evaluation were performed in this simulation environment. The experiments show the efficient performance of the proposed method compared with the other techniques, including SALSA-inspired methods (comparable performance with shorter training time).

**Strengths:**

- Extracting both spatial and speaker information would have important applications in meetings, gaming, and VR scenarios.
- The proposed method shows the efficient performance experimentally.

**Weaknesses:**

- The introduction, related work section, and other sections lack references, making the authors' claims less convincing.
- The paper addresses a narrow topic. Although the challenge of handling short utterances is acknowledged, its practical applications seem limited. While the authors suggest several potential scenarios, the social impacts and practical uses of these approaches remain unclear.
- The paper does not introduce novel machine learning techniques, as it primarily focuses on efficient spatial feature extraction. This limits its appeal to a broader ICLR audience.
- The experimental setup relies entirely on simulations, which may not accurately reflect real-world conditions.
- The simulation methods are based on the authors' in-house implementation, limiting reproducibility and transparency.

**Questions:**

- Generalizability: Can this method be applied to general microphone array scenarios with varying array geometries, including different numbers of microphones?
- Abstract: The term “semantic context” typically doesn’t refer to speaker characteristics. I recommend rephrasing this to clarify that you mean speaker characteristics (e.g., “speaker features”).
- Section 1.1, Last Paragraph: Could you clarify the origin of the 300 ms figure mentioned here?
- Section 2.1, Citation: The current reference, Desplanques et al. (2020), pertains to ECAPA-TDNN. Please cite a more foundational paper for speaker embeddings, such as:
  - Snyder, David, et al. "X-vectors: Robust DNN embeddings for speaker recognition." 2018 IEEE International Conference on Acoustics, Speech and Signal Processing (ICASSP). IEEE, 2018.
- Section 2.1, Speaker Recognition Survey: Consider expanding the literature review on speaker recognition to include additional methods, such as d-vectors and recent advancements in speaker identification.
- Section 3.2, Simulator Details: Please provide more detailed information on the simulator and include relevant references. The current description lacks the necessary detail for reproducibility.
- Section 3.4, Matrix/Vector Processing: The matrix and vector processing in this section is complex. I recommend specifying the domain and the number of dimensions for each variable to clarify the setup.
- Section 3.5, Unitary Matrix Explanation: Could you expand on the phrase “mapped via a unitary matrix, F, to form”? The purpose and nature of the unitary matrix used here are unclear.
- Section 3.5, Properties Proof: I suggest including proofs of the four listed properties in the Appendix to provide further support for these claims.
- Section 3.5, Properties 1 and 2: Adding examples of signals that correspond to Properties 1 and 2 would help readers understand these properties more concretely.
- Experiments in general, Comparison with SOTA: I recommend including comparisons with state-of-the-art speaker ID systems, such as ECAPA-TDNN, to better contextualize your results against established methods.
- Section 4.2, Citation: For the “self-attention pooling Safari et al. (2020)” reference, please consider citing a more foundational paper, such as:
  - Okabe, Koji, Takafumi Koshinaka, and Koichi Shinoda. "Attentive Statistics Pooling for Deep Speaker Embedding." Proc. Interspeech.
- Section 4.2, Explanation of Upstream Training: The statement “we allow our upstream model to continue training while the downstream tasks train” is unclear. Could you elaborate on how the upstream model continues training alongside downstream tasks?
- Section 4.3, VAD Technique: Which voice activity detection (VAD) technique did you use? Please provide details.
- Section 4.3, Noise Distance: The statement, “For the noises, we draw distances further away from the microphone,” is unclear. Could you explain the reasoning behind positioning the noise sources further away?
- Section 4.3, Diffuse Noise Consideration: Did you consider incorporating diffuse noise? Adding this type of noise could make the simulations more realistic.
- Section 4.3, -30dB Noise Level: Using -30dB as a noise level seems unrealistic. Could you confirm whether this level was indeed used, and if so, provide a rationale?
- Section 4.5, Speaker Verification Setup: Consider including a speaker verification experiment with open speaker setups. This addition could make the experiments more generalizable and realistic.
- Section 4.5, Speech-Related Noises in AudioSet: When using AudioSet, did you remove any speech-related noises? Ensuring that speech-related noises are excluded will clarify the task and strengthen the validity of your results.

---

### Official Review · Reviewer_GCFM · 2024-11-05

**Soundness:** 2
**Presentation:** 2
**Contribution:** 2
**Rating:** 3
**Confidence:** 5

**Summary:**

The authors propose a method to identify speakers using short duration speech segments of about 0.3 - 2 seconds, which are very challenging, not just for machines but for humans as well. They propose to leverage spatial information obtained from multi-channel signals to compensate for lack of information in a mono channel  speech signal. There are multiple flaws in the proposed method which are detailed in the weakness section.

**Strengths:**

Identifying speakers using speech segments of very short durations is a challenging task. The proposed method of incorporating speaker location can potentially improve the speaker identification accuracy.

**Weaknesses:**

1. The proposed work focuses on **Speaker Identification using location information**, which is eerily similar to the task of multichannel speaker diarization. Numerous studies have attempted to incorporate location information and speaker embedding to identify speakers. For example:
   - **J. H. M. Wong, I. Abramovski, X. Xiao and Y. Gong**, "Diarisation Using Location Tracking with Agglomerative Clustering," *2022 IEEE Spoken Language Technology Workshop (SLT)*, Doha, Qatar, 2023, pp. 613-619, doi: [10.1109/SLT54892.2023.10022336](https://doi.org/10.1109/SLT54892.2023.10022336).
   - **J. H. M. Wong and Y. Gong**, "Joint Speaker Diarisation and Tracking in Switching State-Space Model," *2022 IEEE Spoken Language Technology Workshop (SLT)*, Doha, Qatar, 2023, pp. 605-612, doi: [10.1109/SLT54892.2023.10023184](https://doi.org/10.1109/SLT54892.2023.10023184).

   Additionally, there are studies that implicitly use location information and train a network to identify the speaker directly from multichannel signals, as proposed in Figure 2 of:
   - **Medennikov, Ivan et al.**, "Target-Speaker Voice Activity Detection: a Novel Approach for Multi-Speaker Diarization in a Dinner Party Scenario," *Interspeech*, 2020.

   The authors should compare their proposed approach in light of these existing works.

2. **Poor Baselines**: The Direction of Arrival (DoA) errors in Table 2 are significantly higher than what is reported for Ambisonic recordings, such as in:
   - **L. Perotin, R. Serizel, E. Vincent, and A. Guérin**, "CRNN-Based Multiple DoA Estimation Using Acoustic Intensity Features for Ambisonics Recordings," *IEEE Journal of Selected Topics in Signal Processing*, vol. 13, no. 1, pp. 22-33, March 2019.

   The authors should provide an explanation for the high DoA errors observed with Learnt_d.

3. **Embeddings Distance**: The primary objective is to identify speakers, with location information serving as supplementary data. I argue that embeddings for the same speaker from different locations should be brought closer together, rather than further apart as proposed in the paper. Since both sounds originate from the same speaker, increasing the distance between their embeddings could compromise the accuracy of speaker identification.

4. **Poor Evaluation Set**: The same set of speakers used in training are reused for evaluation. It is hard to understand the generalization capability of the proposed model.

**Questions:**

Last two paragraphs in Section 4.2 are poorly written and confusing.

---

### Official Review · Reviewer_5XTL · 2024-11-08

**Soundness:** 3
**Presentation:** 3
**Contribution:** 2
**Rating:** 6
**Confidence:** 3

**Summary:**

This paper introduces an interesting task of using spatial information for speaker id. This is an important topic for scenarios like meeting diarization and the proposed methodology is sound. However the innovation in the proposed "POWER VECTOR" is not so sufficient. It is a different covariance mapper which is not necessarily better than other basic mappers by a large margin. Also this paper lacks comparison with other state of art speaker id/diarization methods, so the reviews are not so clear about the real contribution of this paper for real world application, for example, whether this simulated librispeech task really needs spatial information to perform well.

**Strengths:**

Introduced spatial speaker id task is interesting. Simulated dataset is useful.

**Weaknesses:**

The innovation in the proposed "POWER VECTOR" is not so sufficient. It is a different covariance mapper which is not necessarily better than other basic mappers by a large margin. Also this paper lacks comparison with other state of the art speaker id/diarization methods, so the reviews are not so clear about the real contribution of this paper for real world application, for example, whether this simulated librispeech task really needs spatial information to perform well.

Additionally, all the experiments are done only on simulated subset of librispeech which is read speech. The discussion on robustness is not sufficient. Instead of mixed SNRs, the reviews might want to see the performance of the proposed system in different level of SNRs

**Questions:**

Why not comparing to other approaches?

---

### Note · Authors · 2024-11-14

**Comment:**

We have withdrawn our paper as the original authors do not match the authors listed in the submission system.

The original authors are, Dylan Harper-Harris, Richard Cartwright and David McGrath.

**Withdrawal Confirmation:**

I have read and agree with the venue's withdrawal policy on behalf of myself and my co-authors.